# Digital Marketing and Fast-Food Intake in the UAE: The Role of Firm-Generated Content among Adult Consumers

**DOI:** 10.3390/foods12224089

**Published:** 2023-11-10

**Authors:** Ali Ahmed Ali-Alsaadi, L. Javier Cabeza-Ramírez, Luna Sántos-Roldán, Halder Yandry Loor-Zambrano

**Affiliations:** 1Department of Statistics, Econometrics, Operations Research, Business Organization and Applied Economics, Faculty of Law, Business and Economic Sciences, University of Cordoba, 14071 Cordoba, Spain; ali---707@hotmail.com (A.A.A.-A.); luna.santos@uco.es (L.S.-R.); 2Facultad de Ciencias Administrativas y Económicas, Universidad Técnica de Manabí (UTM), Portoviejo 130105, Manabí, Ecuador; halder.loor@utm.edu.ec

**Keywords:** firm-generated content, fast-food consumption patterns, social media engagement, adult consumers, United Arab Emirates, digital marketing, health implications, consumer behavior, online shopping behavior, ethical marketing

## Abstract

In the contemporary digital marketing context, this study aimed to investigate the influence of firm-generated content and social media advertising on fast-food consumption patterns among the adult population. Utilizing a questionnaire distributed to customers of a restaurant in Fujairah, United Arab Emirates, convenience sampling was employed. The findings underscored a significant positive relationship between firm-generated content and social media engagement, as well as between the latter and online shopping behavior. However, it was determined that word of mouth did not significantly moderate the relationship between attitudes towards social media advertisements and fast-food consumption patterns. From a theoretical perspective, these results enrich the understanding of how digital dynamics shape consumer behavior in specific contexts. Practically speaking, they prompt a critical reflection on the ethics of marketing in promoting products potentially detrimental to health, urging both companies and policymakers to reconsider their strategies and regulations, respectively.

## 1. Introduction

In the current society, digitization and the rise of social media have dramatically transformed how businesses engage with their customers and consumers [1,2,3]. In particular, the fast-food industry has exhibited remarkable adaptability to these shifts, strategically incorporating social media platforms as a core component of their marketing strategy [4,5,6]. One of the most notable transitions has been the gradual shift from traditional communication and advertising channels (television, radio, and press) to a new landscape where they coexist with social media advertising [4,7]. Within these platforms, company-generated content plays an essential role in shaping consumer behavior [1,8,9]. This tool serves as direct promotion, facilitating interaction and engagement with the audience, allowing for bidirectional information flows [4,5,10,11,12].

In this context, international fast-food franchises frequently advertise their latest creations and products on these platforms: “Indulge in the Signature Crafted Recipes collection by McDonald’s and discover the sweet and savory flavors from our menu of mouthwatering burgers”; “Enjoy our delicious recipes on single or double 100% fresh beef patties that are sizzled and seasoned on our flat iron grill right when you order” [13]. These campaigns are intensively promoted on the brand’s portal and social media, encompassing text, graphics, photos, videos, or reviews from culinary influencers [1,4,14]. Within hours, the company-generated content becomes available to millions of users [4,9,11]. Any potential social media user can interact with the brand’s post, make an online purchase, try and taste the product, and share their own experience [6,10,15]. For instance, a gourmet burger designed by a renowned chef, accompanied by a special sauce that promises to revolutionize the consumer’s palate [16,17]. This type of advertising strategy, rooted in brand-generated content, is easily replicable for businesses across sizes and sectors [15,18].

Concurrently, political, social, and academic concerns about the effects of exposure to fast-food advertisements on social media have been escalating [19,20,21]. This is, in part, due to repeated warnings from the World Health Organization (WHO) on the health implications of promoting unhealthy diets: “fast food, sugar-sweetened beverages, and chocolate and confectionery” [22]. Recent research has aimed to calibrate these effects, especially among more vulnerable groups like children and adolescents [5,21,23,24,25,26]. However, few studies have focused on the repercussions of advertising on consumption patterns among adults. In this regard, prior investigations like those by Bragg, Pageot, Amico, Miller, Gasbarre, Rummo, and Elbel [4] identified varying interaction behaviors with advertisements based on the target audience. This ties into the potential of fast food (ultra-processed) as a possible trigger for obesity and related health issues [27]. Accordingly, various studies have underscored how cumulative exposure to advertising correlates with fast-food consumption in adults [8,9,28]. Additionally, Vukmirovic [29] highlighted positive associations between advertising, food choices, and consumption patterns [19,20,30]. Pertinent questions arise, such as: does company-generated content influence consumer attitudes toward its social media advertising, their online engagement, and consumption behaviors? More importantly, does the interconnection between variables related to behavior have the capacity to impact fast-food consumption patterns?

Consequently, this study aims to bridge this gap and intends to examine the intricate connections between content produced by fast-food restaurants, consumer attitudes towards this content, and consumption patterns. While there is an extensive body of literature on advertising’s impact on consumer behavior, there is a paucity of research delving into the specific role of restaurant-generated content, especially regarding consumption patterns. Without such studies, it is challenging to understand how social media engagement (SME) or word of mouth (WOM) influence current consumption decisions [31,32]. For this, a new exploratory theoretical model is proposed, grounded in previous findings on social media interactions, attitudes towards digital advertising, and behaviors related to food product selection and consumption. The resulting hypotheses are tested on a sample of 315 customers from a fast-food restaurant in the United Arab Emirates, a context where numerous studies on such establishments are being conducted [8,33,34,35].

## 2. Literature Review and Hypothesis Proposal

Analysis of the effects of advertisements and marketing communications on food and nutrition has become an emerging research line [36,37,38]. Various studies have provided evidence on the potential adverse effects of advertising on the consumption patterns of unhealthy foods [9,23,25,38]. Research focused on fast food faces the absence of a universally accepted definition of the concept [39]. Despite the range of interpretations, for the purposes of this paper, we adhere to the definition provided by Al-Haifi et al. [40], which describes it as a set of foods offered through a limited menu, prepared using production line techniques, served to take away or consume on-site, and focused on products such as burgers, pizzas, chicken, and sandwiches. This definition aligns with the type of products offered at the fast-food restaurant where the surveys were conducted, thus ensuring that perceptions and responses closely align with the study’s central objective.

Within this framework, the impact of advertising on fast food consumption patterns shows significant variations across cultures, influenced by distinct traditions, values, and mindsets [41,42]. While Western societies amplify the promotion of convenience and speed, in cultures with deep culinary traditions, such as Japan or Korea, advertising tends to focus on quality and the fusion of traditional flavors with modern fast-food formats [43,44]. This cultural dichotomy is even more pronounced in the United Arab Emirates, where a massive expatriate population intersects with a rich cultural heritage to create a unique backdrop. Here, the efficacy of social media advertising is shaped by a complexity of sociocultural factors, requiring marketing campaigns to balance universally appealing attributes with local values such as hospitality and family communion during meals [33,34]. Several European studies have shown that food products commonly promoted on television do not adhere to international guidelines (European nutrient profile model of the World Health Organization); for instance, Gallus et al. [45], in a study in Italy, concluded that most food advertisements during children’s viewing times violated these directives. Similarly, in Brazil, based on content analysis of advertisements from different brands associated with fast food, Pereira [46] found the same trend. In New Zealand, Vandevijvere et al. [47] mapped convenience stores, fast food, and takeaway outlets, showing that the country’s schools are surrounded by marketing of unhealthy foods.

In this vein, the troubling connection between fast-food advertising and public health issues like obesity is evident, underlining notable cultural differences [9,48,49,50]. Although advertising for high-energy, low-nutrient foods is associated with an increasing prevalence of obesity across cultures, reactions to these advertising practices vary widely [5,8,49]. In places where obesity is prevalent, fast-food advertising faces scrutiny and stricter regulations are put in place, demanding that companies promote healthy habits and avoid messages that exploit the most vulnerable [38,51]. Conversely, in cultures with traditionally low obesity rates and an emphasis on dietary moderation, advertising has focused less on health and more on the taste and convenience of fast foods [43,44,52]. However, even these societies are not immune to change, as the growing influence of Western lifestyles and the availability of fast food are introducing new public health challenges that demand attention [30,38].

### 2.1. Relationships between Firm-Generated Content (FGC), Attitudes towards Social Media Advertising (ASMA), Social Media Engagement (SME), and Online Shopping Behavior

Kumar, Bezawada, Rishika, Janakiraman, and Kannan [18] conceptualized firm-generated content (FGC) as messages directly emanated by brands on their official platforms and social networks, emphasizing its capability to fortify relationships with customers through the interactive dynamics that social media provides. This variable manifests not only as a conduit offering essential information on products, prices, and promotions, but is also augmented by consumer interactions and evaluations, both positive and negative. In this context, FGC encompasses a variety of content crafted by the brand, including texts, images, videos, and other formats [15,53]. Such content has been shown to have a profound impact in areas like brand recognition, loyalty, and purchase intention [53]. Beyond its intrinsic goals of promotion and engagement [15,53], FGC sways consumer attitudes and values [54], and when paired with positive experiences with products and corporate practices, can result in favorable sentiments [55].

In alignment with this, FGC emerges as a pivotal agent in shaping and adjusting consumer attitudes towards social media advertising (ASMA) [48,53]. This content not only informs but, acting as a paramount source of information, holds the potential to persuade and reshape perceptions [15]. Moreover, due to its ability to incorporate playful and creative elements [34], FGC captures and sustains consumer attention [56]. An illustrative case could be a fast-food restaurant campaign, which, by employing humor and appealing visual design, evokes a more positive response to its social media advertising [57]. This interactive nature of FGC, granting consumers the freedom to express their approval or share content, boosts their engagement [15,53], and could serve as a social endorsement, positively shifting the perceptions of other consumers [58]. Consequently, in juxtaposition with other advertising formats, it may be perceived as more authentic, especially if synergized with user-generated content [15,59].

Based on the above, we propose the following hypothesis: 

**H1.** 
*Firm-generated content (FGC) exerts a positive influence on consumer attitudes towards social media advertising (ASMA).*


Recent studies on FGC have suggested that messages disseminated on company-owned social media platforms have the potential to evoke a positive perception and brand image in consumers [60]. However, they emphasize that further research is still needed regarding the impacts of the two types of FGC (emotional and informational) on consumer engagement behaviors (likes, shares, comments). Social media engagement (SME) signifies the level of commitment and interactions evoked by the brand’s content [8]. It acts as a catalyst by offering relevant, appealing, or emotional content for the consumer [8,60]. Frequently, this FGC is designed to be highly shareable, thus encouraging active user participation. For instance, content that encourages sharing pictures enjoying food at an establishment in exchange for a promotion [15,18,58,59]. Cheng, Liu, Qi, and Wan [60] previously found a relationship between informational and emotional FGC with SME, thereby corroborating earlier findings like those of Pansari and Kumar [61]. Additionally, FGC might include specific calls to action aimed at deeper consumer engagement, such as subscribing to newsletters, taking part in contests, engaging in online communities, and can serve to establish feelings of belonging [15,18,58,59]. Based on the above, the following hypothesis is proposed: 

**H2.** 
*Firm-generated content (FGC) has a positive impact on social media engagement (SME).*


Online shopping behavior refers to the actions that consumers take in the online environment related to the search, selection, purchase, and post-purchase of products or services [62]. This behavior can be influenced by various factors [63,64], including marketing stimuli such as FGC [26]. Given that FGC serves as a primary source of information for the consumer looking to better understand products [61], FGC can enhance the shopping experience by providing a social and emotional context that enriches the consumer’s interaction with the brand [60]. This is especially relevant in the realm of online shopping, where the lack of physical interaction can make consumers feel uncertain [63,64]. Consequently, the following hypothesis is proposed: 

**H3.** 
*Firm-generated content (FGC) has a positive impact on online shopping behavior (OSB).*


### 2.2. Relationships between Attitudes towards Social Media Advertising (ASMA), Social Media Engagement (SME), Online Shopping Behavior, and Fast Food Pattern (FFP)

The consumer’s attitude towards social media advertising (ASMA) not only has the potential to amplify their engagement with certain brands [15,53] but also plays a pivotal role in shaping online purchasing patterns and consumption decisions [49]. This linkage between attitude and engagement is underpinned by the notion that a positive perception of ads can catalyze heightened interaction with the advertising content [48,53]. Furthermore, it has been posited that the attitude towards advertising serves as a significant predictor for both social media participation [65] and online shopping behavior [62,64,66]. In essence, when an individual holds a favorable view of the ads on social media platforms, they are more inclined to interact with brand content [67], which is mirrored in increased digital engagement [68]. For instance, studies have shown that exposure to digital marketing can enhance attitudes and bolster interest in products such as energy drinks [69] and other food items [49]. From a theoretical standpoint, these arguments align with the theory of planned behavior [70] and resonate with the motivations and rewards derived from interacting with advertising [71,72]. In accordance with this theory, positive attitudes towards a product or its advertising often lead to proactive behaviors on social media. Additionally, a favorable attitude towards ads is commonly associated with the perceived utility, entertainment, or informational value they provide [71,72]. In the realm of brand-generated content, a positive attitude towards advertising can manifest in actions like “liking” posts, sharing content, or engaging in brand-related conversations, thereby amplifying the overall engagement rooted in attitudes towards such content [15,18,58,59], and can be an influencing factor in online shopping behavior [63,64,66]. Based on the foregoing, we propose the following hypotheses: 

**H4.** 
*Attitudes towards social media advertising (ASMA) have a positive influence on social media engagement (SME).*


**H5.** 
*Attitudes towards social media advertising (ASMA) have a positive influence on online shopping behavior (OSB).*


The construct Fast Food Intake Pattern (FFIP) refers to the trends and consumption habits associated with fast food. This variable has been less frequently studied in the academic literature as a dependent variable. Understanding it could be crucial to discern how consumer preferences and behaviors translate into specific food choices, or dietary habits, particularly in the context of fast food. Santoso et al. [73], in their work on sodium intake, outline how consumption patterns are defined by repetitive behavior in a given situation. While the development of habits and patterns in daily routines (like eating) optimizes decisions, they might not always lead to positive outcomes if the products consumed are not healthy [5,21,23,24,25,26]. Given this, an individual’s attitude towards social media advertising could influence their fast-food consumption patterns in various ways. A positive attitude towards ads might make the consumer more inclined to try new products or frequent fast-food restaurants more often [1,6,33,35]. This reasoning is grounded in the Theory of Classical Conditioning [74], where repeated exposure to positive stimuli (in this case, attractive ads on social media) can lead to favorable behavioral responses, such as the choice to consume fast food [75]. For instance, if someone sees a social media ad about a new gourmet burger at a fast-food restaurant and has a positive attitude towards that ad, they are more likely to decide to try that burger on their next restaurant visit. This behavior could become a pattern if the individual finds the experience satisfying [73]. Previous studies in marketing and consumer psychology have demonstrated that attitudes towards advertising can influence purchasing decisions and, by extension, consumption patterns [9,36,49,70,76]. Therefore, the hypothesis is proposed:

**H6.** 
*Attitudes towards social media advertising (ASMA) have a positive influence on fast-food patterns (FFPs).*


Online shopping behavior (OSB) encompasses the actions and decisions that consumers make when purchasing products online [63,64]. This variable has been extensively studied in the e-commerce context and has been shown to have a significant impact on various aspects of consumer behavior [77,78]. In this paper, we argue that there may be a relationship between online shopping behaviors and fast-food consumption patterns. The underlying logic of this relationship is that individuals more accustomed to shopping online may be more inclined to use fast food delivery services or mobile apps for ordering [79]. This is based on the Technology Acceptance and Use Theory, suggesting that familiarity with technology and usage behavior facilitate the adoption of similar behaviors (placing orders online) [80], and this acceptance could shape their consumption patterns [73]. For instance, those individuals used to purchase other products online might find it easier and more convenient to use apps to order food from fast-food restaurants, rather than physically visiting the establishment. This behavior might lead to an increase in the frequency with which that person consumes that kind of food, thus establishing a pattern. Furthermore, previous studies have shown that convenience and ease of use are key factors influencing online shopping behavior [64,81]. Based on this, we hypothesize the relationship: 

**H7.** 
*Online shopping behavior (OSB) has a positive influence on fast-food patterns (FFPs).*


### 2.3. The Influence and Moderating Role of Word of Mouth (WOM) on Fast-Food Intake Patterns (FFP)

Word of mouth (WOM) refers to the act of sharing information, opinions, or recommendations about products or services among consumers [82,83]. This phenomenon has been extensively researched in the marketing literature and is deemed one of the most influential methods affecting consumer behavior [84,85].

The linkage between WOM and fast-food consumption patterns (FFPs) is predicated on the notion that recommendations and opinions shared among friends, family, or even influencers concerning brand-generated content can significantly sway an individual’s dietary choices [86,87,88]. This aligns with the Theory of Planned Behavior, suggesting that attitudes and social influences can shape intention and subsequent behavior through subjective norms (consumer’s opinion referents) [89]. For instance, if a close friend positively endorses a newly tried burger from their favorite fast-food restaurant, the likelihood of one being inclined to taste the said food increases. Such endorsements could escalate the frequency of such food consumption, fostering specific consumption patterns [73]. Furthermore, the dissemination of WOM on social media platforms could magnify this effect, given that recommendations and reviews reach a broader audience [85,86]. Based on this understanding, we hypothesize:

**H8.** 
*Word of mouth (WOM) has a positive effect on fast-food intake patterns (FFPs).*


The previous literature suggests that WOM can serve as a moderating factor in various consumer behavior relationships [90,91], including the impact of advertising on purchase decisions [91,92]. In the context of attitudes towards social media advertising (ASMA) and fast-food consumption patterns (FFP), WOM might play a moderating role. Peer or influencer opinions could either bolster or counteract advertising messages, adding another layer of influence on the consumer’s decision-making process. For instance, a consumer might be positively swayed by an advertisement for a new burger at a fast-food restaurant. However, prior to finalizing the purchase decision, they may come across unfavorable online reviews, prompting them to reconsider. This negative WOM could diminish or even negate the initial positive effect that the advertisement had on their attitude [31,84,90]. Therefore, it is plausible to posit that WOM might moderate the relationship between attitudes towards social media advertising and fast-food consumption patterns. This leads us to postulate:

**H9.** 
*Word of mouth (WOM) moderates the relationship between attitudes towards social media advertising (ASMA) and fast-food intake pattern (FFPs).*


Based on the previous review, and the set of hypotheses examined, the resulting model is summarized in Figure 1. 

## 3. Methodology

In accordance with the research objectives, a cross-sectional descriptive–exploratory approach was adopted, particularly well-suited for examining the phenomenon of fast-food consumption in a specific context [93]. This framework allows the analysis of consumer behavior and facilitates the understanding of the relationships between various latent factors, which are usually studied through different measures of influence (i.e., FGC, SME, ASMA, OSB, and WOM). The study was grounded in relationships proposed in prior research pertaining to the constructs included in the preceding model. To achieve this, the partial least squares based structural equation modeling (PLS-SEM) method was employed [93,94]. This approach is particularly suitable when the research objective is aligned with future theoretical developments based on the identified dependency relationships. Specifically, in a reflective–formative configuration, lower-tier constructs are measured in a reflective manner and, while not originating from a shared causal factor, collectively constitute an overarching concept that entirely mediates their impact on subsequent dependent variables [93,95].

### 3.1. Data Collection

The target population for this research consists of adult fast-food restaurant customers in the United Arab Emirates who have some familiarity with content generated by such establishments. For this reason, two initial screening questions were introduced: “Have you consumed fast food at home or at a specialized establishment in the last three months?” and “Have you viewed or received advertising content generated by fast-food restaurants (official website or social media) in the last three months?” No additional restrictions were imposed to ensure randomness in data collection and to fulfill the primary research objective. Following a deductive logic procedure [96], hypotheses were formulated and the supporting literature was reviewed to design a questionnaire based on previously used measurement scales [93].

Data collection was carried out in multiple phases. Initially, a draft questionnaire was prepared and reviewed by five researchers and two fast-food establishment managers. Subsequently, the question order and structure were established, for which 15 interviews were conducted with customers from different establishments. Based on their responses, questions that could induce confusion and potentially bias the results were reformulated. The final instrument was drafted in both English and Arabic using standard bidirectional translation methods. Fieldwork commenced after consultation with the authors’ university ethics committee and was conducted during the months of January to March 2022. Prior to data collection, the establishment manager was informed about the research objectives and permission was sought. Lastly, potential participants were directly contacted during their visits to the establishment. The questionnaire was randomly administered to customers of a restaurant in the city of Fujairah in the United Arab Emirates. Data were collected using convenience sampling [97]. The primary advantage of this sampling method is that it allows the research to focus on the target population [98,99]: fast-food restaurant customers. Those willing to participate were provided with a web link to the electronic-format instrument, informed of the estimated duration (8–15 min), the study objectives, and their participation rights: guarantee of anonymity, confidentiality, and informed consent.

### 3.2. Sample Design and Measurements

The measurement instrument was divided into three sections. The first section included the screening questions, and the second focused on the sociodemographic aspects of the sample, such as age, gender, educational level, employment status, income, and nationality. The third section encompassed the six constructs (variables) that form the theoretical framework of the study (Figure 1). The items constituting each variable were adapted to the research context [62]. Each variable was measured using a 5-point Likert scale (1, strongly disagree; 5, strongly agree). This scale was chosen based on the work of Rehman, Bhatti, Mohamed, and Ayoup [62], who suggested that it enhances the quality and responses by minimizing the level of irritation or frustration caused by 7-point scales. The variable of firm-generated content (FGC) consists of three items adapted from Santiago, Borges-Tiago and Tiago [15], and Kumar, Bezawada, Rishika, Janakiraman, and Kannan [18]. The remaining constructs were also measured with three items: social media engagement (SME) [12]; online shopping behavior (OSB) [62]; attitude towards social media advertising (ASMA) [100,101]; word of mouth (WOM) [31]; and fast-food pattern (FFP) [73]. 

In total, 315 valid responses were obtained, yielding a response rate of 83%. To verify the adequacy of the sample size, G*Power 3.1.9.6 was employed [102]. It was confirmed that the number of valid responses exceeded the predetermined threshold (119 responses). With a statistical power of 0.95, surpassing the 0.8 limit set by Hair, Risher, Sarstedt, and Ringle [93], and the parameters (linear multiple regression: fixed model, R^2^ deviation from zero; effect size f^2^ = 0.15; α err prob = 0.05; power (1-β err prob) = 0.95; number of predictors = 3), questionnaires with missing values or those that did not affirmatively answer both screening questions were excluded. Table 1 presents the primary sociodemographic data of the study. Most restaurant visitors are of Emirati origin (88.4%). The gender distribution leans slightly in favor of the male population (53.7% versus 46.3%). Just over half of the respondents are employed (53%). Regarding age, the most represented group is individuals between 25 and 34 years old (32.7%), followed by those in the 35 to 44 age bracket (31.4%). Most respondents have higher education or university degrees and moderate income levels.

### 3.3. Data Analysis

The analysis of the proposed model was conducted using the partial least squares (PLS) variance-based technique [103]. The implementation of this technique followed the guidelines set forth by Hair and Alamer [104]. Specifically, SmartPLS software (version 4) was utilized for the analysis [105]. The relationships between various indicators and constructs were examined using a latent variable modeling approach [103]. As is customary in studies employing variance-based structural equation models, both the measurement model and the structural model were validated [106]. All latent variables in the proposed model are composite factors formatted in A-mode [107]. This presupposes that the variables consist of an aggregate of multiple indicators, which, when measured in A-mode, are permitted to correlate with one another [108]. 

## 4. Results

The validation of the measurement model for the latent variables modeled in A-mode consists of several stages: estimating the individual reliability of each item in each construct, measuring the reliability of each latent variable, determining the convergent validity of the construct, and, finally, assessing the discriminant validity of the constructs [107,108]. The individual reliability of each item is performed by examining the loadings (λ) (simple correlations) of each indicator with the construct to which it belongs. The criteria established in the literature set the critical level at a value of λ ≥ 0.707 [109]. The results corresponding to our model are presented in Table 2 and show that of the six constructs, only one of them loses an element—“Fast-Food Pattern”—where the item “I love fast-food and snacks” must be eliminated.

The internal consistency (or reliability) of a latent variable makes it possible to establish the degree of rigor with which the observed (manifested) variables measure a given construct. Three measures are commonly used in PLS to determine the reliability of a construct: Cronbach’s alpha coefficient, Werts’ composite reliability (ρc) [110], and Dijkstra-Henseler’s rho A (ρA) [111]. The rules or criteria state that the minimum threshold for these three indicators should be equal to or greater than 0.8. Table 2 shows the composite reliability values for the six variables in our model, with all exceeding the minimum threshold. To determine the convergent validity of an unobserved variable (construct), it is customary to use a single measure: the average variance extracted [85,112]. The aim is to evaluate the amount of variance that a construct receives from its indicators in relation to the quantity of variance due to measurement error. The cut-off point is set at an AVE of 0.5 or greater. All six constructs in this model, as Table 2 reflects, far exceed that cut-off point.

The last aspect to consider in the case of latent variables estimated in A-mode is discriminant validity. This involves establishing the extent to which a construct differs from the rest. The methods used to assess this discriminant validity are cross-loading analysis, the Fornell and Larcker [112] method, and the heterotrait–monotrait ratio (HTMT). Given that the latter is the most stringent among them, it is the one we have employed in our study. The HTMT ratio was proposed by Henseler et al. [113] and involves the comparison of heterotrait correlations with monotrait correlations. The criteria for determining the existence of discriminant validity point to the following thresholds: HTMT ratio < 0.85 [114] or HTMT ratio < 0.9 [115]. To test whether the HTMT ratio is significantly different from 1, the bootstrapping technique can be used: if the confidence interval for the HTMT ratio includes the value 1, discriminant validity cannot be confirmed; otherwise, discriminant validity can be affirmed. The results of this ratio are shown in Table 3 and confirm the existence of discriminant validity, in line with the guidelines set forth by Gold, Malhotra, and Segars [115].

Once the validity of the measurement model has been ensured, the evaluation of the structural model can proceed. To do so, we have adhered to the guidelines outlined in the literature [109]. First, we proceeded to check for the absence of multicollinearity between the antecedent variables of each endogenous construct. According to Hair Jr, Hair Jr, Hult, Ringle, and Sarstedt [109], the absence of multicollinearity is confirmed when the values of the variance inflation factor (VIF) indicator are less than 5. The VIF values obtained for the various relationships were as follows: FGC–SME (1.483), FGC–ASMA [52], FGC–OSB (1.483), ASMA–SME (1.483), ASMA–OSB (1.483), ASMA–FFP (4.068), WOM–FFP (3.887), and OSB–FFP (3.589).

Figure 2 presents the estimates of the structural model. This illustration underscores the robust explanatory power of the proposed model. Specifically, three of the endogenous variables—fast-food intake pattern, online shopping behavior, and social media engagement—exhibit R^2^ values that are considered high, exceeding 0.6. Meanwhile, the R^2^ value for the fourth endogenous variable, attitudes towards social media advertising, falls within a moderate range [93].

The results corresponding to the path coefficients (sign, size, and significance), the Q^2^ test, and the values of the coefficient of determination (R^2^) are given below. For the testing of the hypotheses suggested in the model and the corresponding assessment of their significance and relevance, we proceeded to use the bootstrapping technique (with 5000 subsamples), according to the indications of Hair Jr, Hair Jr, Hult, Ringle, and Sarstedt [109]. The results of this analysis are provided in Table 4 and confirm that the eight direct hypotheses indicated in the model are confirmed.

The results of hypothesis 1 (β = 0.571, *p* = 0.000) show that the firm-generated content positively condition consumers’ attitudes toward advertisements in social media. On the other hand, the hypotheses referring to consumers’ social media engagement are confirmed: highlighting that the effect of firm-generated content (H2, β = 0.604, *p* = 0.000) is higher than the direct effect of attitudes toward ads in social media (H4, β = 0.290, *p* = 0.000). The hypotheses related to consumers’ online shopping behavior also turn out to be significant and positive. Thus, it is found that, in this case, the influence of attitudes toward advertisements on social media is more important (H5, β = 0.676, *p* = 0.000) than the firm-generated content (H3, β = 0.249, *p* = 0.000). The results regarding consumers’ fast-food intake pattern show that all the variables have positive and significant effects. The most significant effect corresponds to the variable “attitude towards social media advertising” (H6, β = 0.381, *p* = 0.000). The other two variables show practically similar effects, “WOM” (H8, β = 0.247, *p* = 0.000) and “online shopping behavior” (H7, β = 0.230, *p* = 0.000).

Regarding hypothesis 9 (β = −0.028. *p* = 0.131), which indicates the moderating effect of word-of-mouth comments on the relationship between “attitudes towards social media advertising” and “fast-food intake patterns”, the results show the absence of a significant relationship. In other words, word-of-mouth comments do not significantly condition this relationship. The coefficient of determination (R^2^) allows us to measure the predictive power of any model and, on the other hand, indicates the amount of variation of a construct that is explained by its predictor variables (Table 5). According to [109] and reorder the ref “Hair et al. (2017)”, our model can explain the variable “fast-food intake pattern” significantly, to a moderate degree the constructs of “social media engagement” and “online shopping behavior” and, finally, weakly in the case of the variable “attitude towards social media advertising”.

In addition, the preceding table also highlights that all the variables in the model have sufficient explanatory power given the levels of effect sizes, based on Cohen [116] f^2^ statistic values. The model also offers meaningful significance values, as all the constructs reach positive Q^2^ values well above 0. Although in our model we have not proposed hypotheses on mediated relationships between the variables, we consider it appropriate to comment briefly on the results shown in Table 6. The results presented in this table show that all the indirect paths present positive and significant values and that all the mediations are complementary in nature.

## 5. Discussion and Conclusions

The primary purpose of this research was to delve deeper into understanding the interaction between the content generated by fast-food companies, specifically through social media advertising, and its influence on fast-food consumption patterns. This investigation emerged from the need to bridge the existing gap in the literature regarding how new forms of communication and promotion, in an increasingly digitalized world, affect consumption habits, specifically among the clientele of these establishments.

From the results obtained and visualized in the presented model, there was a significant relationship between the company-generated content (FGC) and the attitude towards social media advertising (ASMA). This relationship reinforces the idea that brand information, when effectively presented, can positively influence consumer perception of advertising [48,53]. The connection between WOM and fast-food consumption patterns (FFP) was also significant. It was found that recommendations and opinions shared among individuals have a substantial impact on the possible formation of consumption patterns [86,87,88]. This finding aligns with previous research emphasizing the influence of WOM on consumer behavior [84,85]. Another notable outcome was the positive relationship found between the attitude towards social media advertising (ASMA) and online purchasing behavior (OSB). This suggests that the perception of social media advertising might lead to an inclination towards online purchasing, consistent with current market trends [63,64,66]. Conversely, it is essential to note that WOM showed no moderation in the relationship between ASMA and FFP. Although this outcome might seem surprising, it aligns with some studies suggesting that WOM does not always moderate relationships between variables, specifically in fast-food research [96]. This lack of moderation could indicate the prevalence of other influential factors not explored in this study, highlighting the complexity of consumer behavior in today’s digital environment. The absence of moderation effects in the findings, particularly between attitudes toward social media advertising (ASMA) and fast-food consumption patterns (FFP), could be attributed to multiple intrinsic and extrinsic factors that affect consumption decisions [117,118]. It is possible that individual consumer characteristics, such as personal values, the need for belonging, or their level of involvement with the product category [119,120,121], play a more critical role in how they perceive and respond to social media advertising [122]. Furthermore, the ubiquity of digital advertising could have led to saturation [123], diminishing the ability of WOM to further influence established consumption patterns. In this context, the impact of message credibility and source must also be considered. WOM, traditionally seen as a reliable source of information due to its organic nature and peer trust, may be competing with a media landscape where influencers and sponsored content creators have begun to blur the lines between user-generated content and advertising. This phenomenon could have weakened the moderating influence of WOM on the relationship between ASMA and FFP. These aspects underscore the need for deeper research to unravel the dynamics of moderating influences in advertising and fast-food consumption. The integration of variables such as source credibility, advertising saturation, and individual consumer characteristics could provide a richer and more nuanced understanding of these relationships [83,85,90,91]. Exploring these dimensions would enable researchers and marketing professionals to develop more effective strategies that align with the increasingly complex consumer behavior in the constantly evolving digital ecosystem.

Given the growing global concern over health issues associated with excessive fast-food consumption, such as obesity and heart diseases, it is crucial to understand the factors driving these consumption patterns in adults [30,36,38,49,73]. The evidence presented in this research indicates that the digital strategies adopted by fast-food companies, especially on social media, have a tangible influence on consumer perception and behavior. It is imperative that authorities and stakeholders consider these findings when formulating policies or interventions aimed at mitigating the negative impact of fast food on public health. In a world where digitalization continues to transform consumer interactions, recognizing and addressing the challenges presented by the confluence of technology and food is essential to ensure the well-being of current and future generations. The increase in fast-food consumption, influenced by advertising on social media, is associated with various negative health consequences that require urgent attention [124]. Scientific evidence has shown a correlation between frequent fast-food consumption and the rise of non-communicable diseases, such as obesity, high blood pressure, type 2 diabetes, and other metabolic disorders [30,36]. These diseases represent a growing challenge for global public health and entail high economic and social costs, exacerbated by overburdened health systems and the loss of labor productivity [125]. In this scenario, public health policies must focus on the treatment of these conditions and on their prevention through effective intervention strategies [126]. Such strategies could include the implementation of taxes on high-caloric, low-nutritional value foods, restrictions on the timing of fast-food advertising aimed at minors, and awareness campaigns about the health risks of excessive consumption of these products [51,127,128]. In addition, policies could promote healthier food environments through tax incentives for restaurants that offer healthier options and the creation of fast-food-free zones near schools [129].

Furthermore, considering the central role that social media plays in shaping consumption habits, it is crucial for digital platforms to actively participate in promoting healthy lifestyles [1,34]. This could be achieved through algorithms that favor content related to healthy nutrition and physical activity [130], as well as through collaboration with health authorities to spread prevention messages [131]. Lastly, the regulation of advertising messages on social media should include a clear presentation of nutritional information and warnings about the risks associated with the consumption of the promoted products [25,50,132]. If these measures are implemented comprehensively and accompanied by continuous research, they have the potential to modify dietary consumption patterns and can contribute to improving health and reducing the incidence of diet-related diseases [73].

### 5.1. Theoretical Implications

From a theoretical perspective, this research sheds light on the comprehensive influence that firm-generated content by fast-food brands has on consumption patterns, especially within an adult demographic. Traditionally, theories on consumer behavior have focused on more tangible factors such as price, product quality, and location. However, in today’s digital age, where brand–consumer interaction has evolved to be more immersive and personalized thanks to social media, it is evident that traditional dynamics have shifted. The discovery that firm-generated content can significantly influence consumer attitudes toward advertising, and more importantly, their consumption patterns, suggests a need to re-evaluate and expand existing theories. Specifically, the relationship between firm content and consumption patterns may be mediated by multiple factors, including the perceived authenticity of the content, exposure frequency, and the emotional nature of the content.

Secondly, this work reveals that consumption patterns are being shaped both by direct advertising and by word of mouth (WOM). Although the moderation of WOM in the relationship between attitudes towards advertising and consumption patterns was not significant in this research, the fact that WOM itself impacts consumption patterns underscores the importance of social media and online communities in shaping consumption decisions. This research highlights the need for a theoretical review that integrates the complexities of digital interaction and its influence on consumption patterns. Traditional models need to be adapted or expanded to encompass the wide range of stimuli and responses that define the relationship between fast-food brands and their consumers in the digital world.

### 5.2. Practical Implications

The connection between digital marketing and consumption behavior in the fast-food industry among the adult population provides a series of essential practical implications for marketing professionals and companies in the sector. Given that this study focuses on the adult population, which often has a higher purchasing power and more autonomous purchase decisions compared to younger populations, it is vital that companies understand how to influence this demographic segment responsibly. In relation to firm-generated content (FGC), it is essential that brands invest in creating genuine, relevant, and attractive content that resonates with the adult population, warning of harmful consumption patterns. Strategies that go beyond simple promotions and focus on values, healthy lifestyles, and social responsibility might have a more profound impact on this demographic group.

Furthermore, given the potential health repercussions associated with frequent fast-food consumption, companies have an ethical responsibility. They should be transparent in their advertising practices and consider promoting healthier options on their menus, aligning with society’s growing demands for more conscious and healthier eating. Lastly, although WOM did not show a significant moderating effect between ASMA and FFP, brands should not underestimate the power of online opinions. They should actively monitor and respond to reviews and comments on digital platforms, ensuring that the brand’s perception remains positive and that potential issues are addressed swiftly. To translate ethical responsibility into concrete actions, fast-food companies can adopt a multifaceted approach to their marketing and advertising strategies. In line with the findings of Otto, Johnston, and Baumann [16], advertising transparency can be improved by using clear and visible labels that detail the nutritional information of products, enabling consumers to make informed decisions [133]. For instance, incorporating QR codes on packaging that link to detailed information about calories, ingredients, and healthy options can encourage conscious choice. Regarding the promotion of healthier choices, companies can highlight these alternatives on their menus using “choice architecture”. Positioning healthy options in prominent places both on physical menus and digital platforms can positively influence consumer decisions [134]. Additionally, including loyalty programs that reward the selection of healthy options with discounts or additional benefits could motivate a change in consumption habits [135]. On the other hand, regarding the creation of brand content, the focus should be on campaigns that highlight healthy lifestyles [136]. Collaborations with influencers or personalities who promote good nutrition, and an active lifestyle can have a significant impact. These campaigns could include healthy cooking challenges or physical activity competitions, using digital platforms to generate engagement and awareness.

### 5.3. Limitations and Future Lines of Research

This study, like any empirical research, has certain limitations that must be acknowledged. Firstly, the sampling method used, although offering valuable insights into the target audience of fast-food restaurant customers, has a significant shortcoming. Despite its advantage in focusing on a specific population, it may not be representative of the broader universe of fast-food consumers. Measures have been taken to ensure that the sample reflects relevant characteristics of the population under study (fast-food restaurant customers in the United Arab Emirates); however, the results should be interpreted with the understanding that they represent the trends and behaviors of the study participants, not necessarily all fast-food consumers. This limitation is counterbalanced by the detailed analysis and contextual relevance provided for stakeholders and professionals in the fields of marketing and consumer behavior within the region being examined. As Baltes and Ralph [137] suggest, the aim of this study is not to generalize its findings to all populations and regions but rather to offer a rigorous analysis of the sample, delivering a detailed and contextualized understanding of consumer behavior within this specific segment. Secondly, the research was conducted in the city of Fujairah in the United Arab Emirates, which might not mirror trends and behaviors in other regions or cultures. As such, caution should be exercised when generalizing the findings to a broader context. Moreover, even though the questionnaire was distributed randomly, providing an electronic link for its completion means that those customers without access to or unfamiliar with digital technology were excluded. This could have skewed the results towards a younger or more technologically advanced demographic. Thirdly, the measurement scales used were limited; the pattern of fast-food consumption, while vital for this study, was measured using only two items. This might not capture the complexity and depth of the construct in its entirety. Future research could benefit from including additional items or employing more robust scales to assess this construct more accurately. Fourthly, the study’s focus on the impact of social media advertising might overshadow the potential influence of other advertising channels, both digital and traditional, in shaping fast-food consumption patterns [50]. Traditional media, such as television, radio, and print, as well as other digital platforms beyond social media, may also play a significant role in consumer behavior and choices [38,138].

These limitations present opportunities for future research. Replicating this study in various geographical and cultural contexts would be invaluable to better understand the universality or specificity of these findings. Furthermore, delving deeper into the pattern of fast-food consumption with more detailed measures could shed light on other influencing factors. It would also be relevant to investigate consumer resistance or awareness concerning digital marketing influences, given the unique non-positive relationship found in this study. With these considerations in mind, this research establishes a starting point for future investigations in the field of digital marketing, consumer behavior, and health impacts in the context of the fast-food industry. Additionally, future studies could expand the scope to include traditional media channels to provide a more comprehensive view of the multifaceted advertising landscape and its effects on fast-food consumption.

## Figures and Tables

**Figure 1 foods-12-04089-f001:**
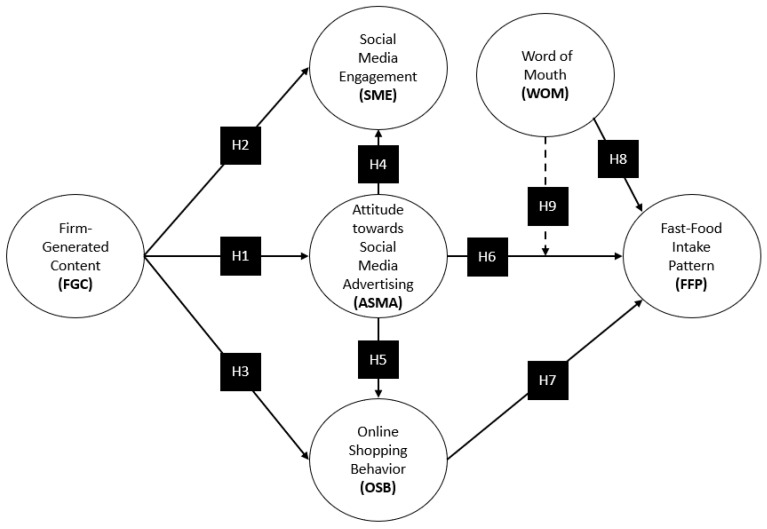
Research model.

**Figure 2 foods-12-04089-f002:**
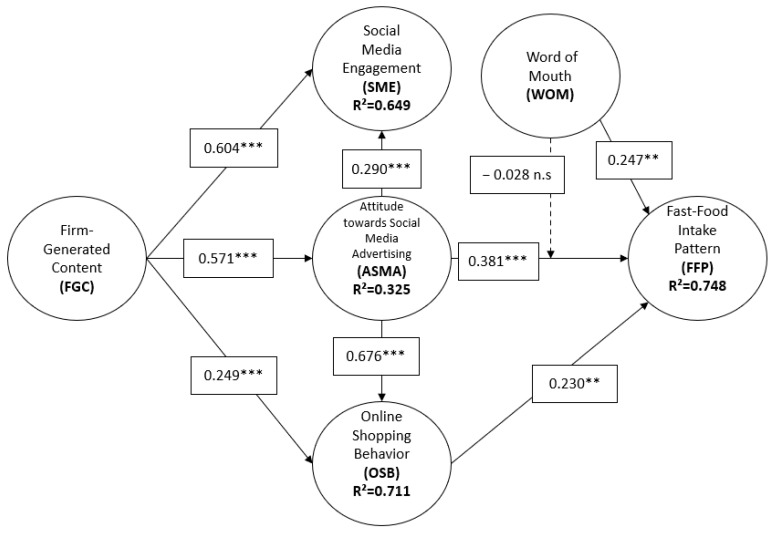
Model estimation. *** *p* < 0.001, ** *p* < 0.01; ns = not significant.

**Table 1 foods-12-04089-t001:** Sample characteristics.

Criteria	Levels	%
Gender	Men	53.7
	Women	46.3
Age	18–24	29.4
	25–34	32.7
	35–44	31.4
	45–65	4.9
	More than 65	1.6
Level of studies	Preparatory	7.9
	High school	34.7
	University	57.1
	Other	0.3
Laboral activity	Student	22.1
	Self-employed	13.4
	Employed	53
	Unemployed	11.5
Income level [Arab Emirates Dirham (AED) by year]	1000–4999	24.8
	5000–9999	14.2
	10,000–19,999	15.2
	20,000–29,999	25.4
	30,000–40,000	15.8
	More 40,000	4.6
Nationality	United Arab Emirates	88.4
	Other	11.6

**Table 2 foods-12-04089-t002:** Reliability measures.

Constructs/Items	Loading (λ)	Composite Reliability	AVE	Scale Adapted from
**Firm-generated content (FGC)**		**0.981**	**0.944**	[15,18]
**FGC1.** The content generated by the company (web, social media) about its products is very attractive	0.970			
**FGC2.** The content generated by the company (web, social media) meets my expectations	0.970			
**FGC3.** The content generated by the company (web, social media) satisfies me	0.975			
**Social media engagement (SME)**		**0.970**	**0.916**	[12]
**SME1.** I participate in the restaurant’s social media through “like”, “comment”, and “share”	0.957			
**SME2.** I like to participate in the social media of the fast-food restaurant	0.948			
**SME3.** I interact with the social media of the fast- food restaurant	0.965			
**Online shopping behavior (OSB)**		**0.959**	**0.886**	[62]
**OSB1.** I frequently buy fast food online because it is convenient for me	0.937			
**OSB2.** I consider online shopping for fast food to be compatible with my lifestyle	0.945			
**OSB2.** Online shopping simplifies my day-to-day purchases, especially food	0.942			
**Attitude towards social media advertising (ASMA)**		**0.956**	**0.880**	[100,101]
**ASMA1.** In your opinion, social media advertisements are entertaining	0.956			
**ASMA2.** In your opinion, social media advertisements are annoying (Reversed question)	0.915			
**ASMA3.** In your opinion, social media advertisements are useful	0.942			
**Word of mouth (WOM)**		**0.977**	**0.934**	[31]
**WOM1.** My friends and family make recommendations about this fast-food restaurant.	0.970			
**WOM2.** They have given me positive comments about this fast-food restaurant	0.965			
**WOM3.** They have told me about their experience with this fast-food restaurant	0.964			
**Fast-food pattern (FFP)**		**0.958**	**0.920**	[73]
**FFP2**. Eating fast-food is an act I do without thinking	0.961			
**FFP3.** I find it very difficult to avoid fast-food	0.958			

**Table 3 foods-12-04089-t003:** Discriminant validity (HTMT ratio).

	ASMA	FGC	WOM	WOM ASMA	SME	OSB
FGC	0.598					
WOM	0.877	0.693				
WOM ASMA	0.780	0.451	0.837			
SME	0.671	0.799	0.756	0.521		
OSB	0.873	0.664	0.848	0.658	0.745	
FFP	0.898	0.594	0.861	0.741	0.682	0.849

**Table 4 foods-12-04089-t004:** Hypotheses test.

Hypotheses	Suggested	Path	T Value	Confidence Interval	
	Effect	Coefficient (β)			
				5.0%	95.0%
H1: FGC **→** ASMA	(+)	0.571 ***	11.295	0.482	0.649 Sig
H2: FGC **→** SME	(+)	0.604 ***	13.159	0.526	0.677 Sig
H3: FGC **→** OSB	(+)	0.249 ***	4.923	0.165	0.332 Sig
H4: ASMA → SME	(+)	0.290 ***	6.495	0.218	0.363 Sig
H5: ASMA **→** OSB	(+)	0.676 ***	14.878	0.600	0.752 Sig
H6: ASMA **→** FFP	(+)	0.381 ***	3.903	0.218	0.540 Sig
H7: OSB **→** FFP	(+)	0.230 **	2.635	0.085	0.372 Sig
H8: WOM → FFP	(+)	0.247 **	2.497	0.084	0.408 Sig
H9: WOM ASMA → FFP	(+)	−0.028 ns	1.122	−0.069	0.012

*** *p* < 0.001, ** *p* < 0.01, t (0.05; 4999) = 1.64791345; t (0.01; 4999) = 2.333843952; t (0.001; 4999) = 3.106644601. Sig. denotes a significant direct effect at 0.05; ns = not significant.

**Table 5 foods-12-04089-t005:** Effect on the endogenous variables.

	Adjusted R^2^	Q^2^	Direct Effect	Correlation	Variance	Effect Size (f^2^)
					Explained	
**ASMA**	0.323	0.272				
H1: FGC			0.571	0.571	32.6%	0.483
**SME**	0.647	0.585				
H2: FGC			0.604	0.770	46.5%	0.702
H4: ASMA			0.290	0.635	18.4%	0.162
**OSB**	0.709	0.621				
H3: FGC			0.249	0.634	15.8%	0.144
H5: ASMA			0.676	0.818	55.3%	1.066
**FFP**	0.744	0.666				
H7: OSB			0.230	0.787	18.1%	0.056
H8: WOM			0.247	0.809	20.0%	0.043
H6: ASMA			0.381	0.829	31.6%	0.130
H9: WOM ASMA			−0.028	−0.709	2.00%	0.006

**Table 6 foods-12-04089-t006:** Summary of mediating effects.

	Coefficient	Bootstrap 95% CI			
	Point Estimate	Percentile		BC	
**Individual indirect effects**					
FGC -> ASMA -> SME	0.166 ***	0.115	0.223	0.117	0.226
FGC -> ASMA -> OSB	0.386 ***	0.307	0.465	0.308	0.466
FGC-> ASMA -> FFP	0.217 ***	0.121	0.317	0.124	0.320
FGC -> OSB -> FFP	0.057 **	0.021	0.096	0.024	0.101
FGC -> ASMA -> OSB ->FFP	0.089 **	0.031	0.153	0.035	0.158
ASMA -> OSB -> FFP	0.156 **	0.056	0.259	0.059	0.260
**Total indirect effect**					
(FGC -> SME)	0.166 ***	0.115	0.223	0.117	0.226
(FGC -> OSB)	0.386 ***	0.307	0.465	0.308	0.466
(FGC -> FFP)	0.363 ***	0.273	0.455	0.276	0.458
(ASMA -> FFP)	0.156 **	0.056	0.259	0.059	0.260

*** *p* < 0.001, ** *p* < 0.01.

## Data Availability

The data used to support the findings of this study are available from the first author upon request.

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
