# Peer review of "Digital Marketing and Fast-Food Intake in the UAE: The Role of Firm-Generated Content among Adult Consumers"

_foods, 2023, doi:10.3390/foods12224089_

Round 1

Reviewer 1 Report

Comments and Suggestions for Authors

1. In my opinion, the general literature on PLS as a methodology is unnecessary, since it is known and the article is not a theoretical study

2. It is desirable to change and expand the review from the point of view of research on such issues in the world, including taking into account differences in traditions and mentality

Author Response

First and foremost, we would like to express our sincere gratitude for the insightful comments and suggestions provided, as well as for the valuable time invested in reviewing our manuscript.

Comments and Suggestions for Authors

Author's Reply 

In my opinion, the general literature on PLS as a methodology is unnecessary, since it is known, and the article is not a theoretical study

We appreciate the reviewer's point regarding the inclusion of general PLS literature in our manuscript. While we recognize that the article is not a theoretical study, and the special issue on consumer behavior will predominantly be read by researchers accustomed to this methodology, we deemed it appropriate to retain references to PLS. This consideration stems from the fact that the journal 'Foods' has a broad readership and some readers from the food sector may not be fully versed in PLS methodology. Therefore, we believe that providing this information is valuable in ensuring the study's comprehension and accessibility to a diverse audience.

It is desirable to change and expand the review from the point of view of research on such issues in the world, including taking into account differences in traditions and mentality

Thank you for your comment. We have increased the literature review section (lines 92 to 121), and made changes to the discussion and conclusions, as well as to the implications of the paper by addressing differences in traditions and culture.

Reviewer 2 Report

Comments and Suggestions for Authors

The papae ris interesting and valuable but there are some concerns needs atention:

-The research acknowledges that the sampling method used may not be fully representative of the broader universe of fast-food consumers. This limitation can raise concerns about the generalizability of the findings to different populations or regions.

-Although the paper highlights the need for fast-food companies to be transparent in their advertising practices and promote healthier menu options, it does not delve into the specifics of how companies can achieve this. Practical guidance on implementing these ethical changes is lacking.

-While the paper focuses on the impact of social media advertising, it may understate the potential role of other advertising channels, both digital and traditional, in influencing fast-food consumption patterns.

-Provide a detailed explanation for non-significant findings, especially in cases where moderation effects are not observed. This can help readers understand the nuances of the relationships being explored.

-Deepen the discussion of the health implications of fast-food consumption. Explore the specific health consequences and policy implications in greater detail. Consider the potential for interventions to mitigate health risks.

-Include a section in the paper discussing potential biases in the research design, data collection, and analysis. This transparency will enhance the credibility of the study.

-Conclude the paper with a section outlining potential directions for future research. This can guide other researchers in building upon your work.

-Improve the presentation of data visualization, particularly figures and tables, to make the results more accessible and understandable to readers.

Comments on the Quality of English Language

It's ok.

Author Response

First and foremost, we would like to express our sincere gratitude for the insightful comments and suggestions provided, as well as for the valuable time invested in reviewing our manuscript.

Comments and Suggestions for Authors

Author's Reply 

-The research acknowledges that the sampling method used may not be fully representative of the broader universe of fast-food consumers. This limitation can raise concerns about the generalizability of the findings to different populations or regions.

We appreciate your insightful comment. Based on it, we have specified in the study limitations (lines 650 to 660) the appropriate caveats regarding the generalization of results.

-Although the paper highlights the need for fast-food companies to be transparent in their advertising practices and promote healthier menu options, it does not delve into the specifics of how companies can achieve this. Practical guidance on implementing these ethical changes is lacking.

We appreciate your comment, it is a pertinent suggestion. After a re-reading of the practical implications of the work we have implemented practical guidelines (between lines 627 to 644)

-While the paper focuses on the impact of social media advertising, it may understate the potential role of other advertising channels, both digital and traditional, in influencing fast-food consumption patterns.

We believe that this is a good recommendation, and we have included it as the fourth limitation of the work (lines 670 to 675), which in turn is a good line for future work (lines 684 to 686).

-Provide a detailed explanation for non-significant findings, especially in cases where moderation effects are not observed. This can help readers understand the nuances of the relationships being explored.

Thank you for your comment. We have tried to provide the reader with one of the possible explanations why no moderator effect was observed in this case (lines 529 to 549).

-Deepen the discussion of the health implications of fast-food consumption. Explore the specific health consequences and policy implications in greater detail. Consider the potential for interventions to mitigate health risks.

Thank you for your comment, we have incorporated an extension of the discussion between lines 559 to 584.

-Include a section in the paper discussing potential biases in the research design, data collection, and analysis. This transparency will enhance the credibility of the study.

Thank you for your comment, we believe that with the information provided in section 3.1 Data Collection, 3.2 Sample Design and measurements, together with the comments added in the limitations this question is answered.

-Conclude the paper with a section outlining potential directions for future research. This can guide other researchers in building upon your work.

We have expanded the future lines of research included (lines 676 to 686).

-Improve the presentation of data visualization, particularly figures and tables, to make the results more accessible and understandable to readers.

We have improved the display of figures and tables in response to your request. (See figure 1 and 2)
